# Suppression of Tomato Bacterial Wilt Incited by *Ralstonia pseudosolanacearum* Using Polyketide Antibiotic-Producing *Bacillus* spp. Isolated from Rhizospheric Soil

Dinesh Singh [1,*], Venkatappa Devappa [2,*] and Dhananjay Kumar Yadav [1]

1 Division of Plant Pathology, Indian Council of Agricultural Research-Indian Agricultural Research Institute, New Delhi 110012, India
2 Department of Plant Pathology, College of Horticulture, University of Horticultural Sciences, GKVK Post, Bengaluru 560065, India
* Correspondence: dinesh_iari@rediffmail.com (D.S.); devappav@gmail.com (V.D.)

**Abstract:** *Bacillus* spp. has the potential to control bacterial and fungal diseases of crops. In vitro study, *Bacillus amyloliquefaciens* DSBA-11 showed best to inhibit the growth of *Ralstonia pseudosolanacearum* as compared to *Bacillus cereus* JHTBS-7, *B. pumilus* MTCC-7092, *B. subtilis* DTBS-5 and *B. licheniformis* DTBL-6. Three primers sets from nucleotide sequences of polyketide antibiotic synthase genes *viz.*, macrolactin, difficidin and bacillaene of *B. amyloliquefaciens* FZB42 were designed and standardized protocol for simultaneous detection of polyketide antibiotics-producing strains of *Bacillus* spp. by multiplex—PCR with products size of 792 bp, 705 bp and 616 bp respectively. All the strains of *B. amyloliquefaciens* contained three polyketide antibiotic synthase genes, and *B. subtilis* possessed difficidin and macrolactin, whereas *B. cereus* JHTBS-7, *B. pumilus* MTCC-7092 and *B. licheniformis* DTBL-6 did not contain any polyketide antibiotic genes. By using this technique, polyketide-producing strains of *Bacillus* spp. were screened within a short period with high accuracy. These polyketide synthase genes were cloned by using a T&A vector to study the role of these genes in producing antibiotics that suppressed the growth of *R. pseudosolanacearum* under both in vitro and in vivo conditions. Bio-efficacy of cloned products of these genes macrolactin, bacillaene, and difficidin along with parent strain *B. amyloliquefaciens* DSBA-11 inhibited the growth of *R. pseudosolanacearum* and formed 1.9 cm$^2$, 1.9 cm$^2$, 1.7 cm$^2$ and 3.3 cm$^2$ inhibition area under in vitro conditions respectively. Minimum bacterial wilt disease intensity (29.3%) with the highest biocontrol efficacy (57.72%) was found in tomato cv. Pusa Ruby (susceptible to wilt disease) was treated with *B. amyloliquefaciens* DSBA-11 followed by cloned products of difficidin and macrolactin under glasshouse conditions. Hence, the developed multiplex protocol might be helpful for screening polyketide antibiotics producing potential strains of *Bacillus* spp. from soil which can apply for managing the wilt disease of tomatoes. The polyketide antibiotics produced by bacteria might have a significant role suppression of *R. pseudosolanacearum* due to the disintegration of cells.

**Keywords:** antagonistic; bacterial wilt; multiplex PCR; polyketides antibiotics; gene cloning; *Bacillus* spp.; *Ralstoniap seudosolanacearum*; tomato

## 1. Introduction

Bacterial wilt caused by *Ralstonia pseudosolanacearum* (Formely *Ralstonia solanacearum* (Smith) Yabuuchi (1995) [1] is a serious disease of tomato (*Solanum lycopersicum* L.) in tropical, subtropical and temperature areas of the world. In India, the bacterial wilt disease has been reported mostly in coastal, hilly, as well as foothills areas, in India including in the state of Goa, Karnataka, Kerala, Maharashtra, Odisha, Jharkhand, and West Bengal, Himachal Pradesh, Jammu & Kashmir, Uttarakhand and north-eastern states [2,3]. The disease causes very heavy losses, varying from 2 to 90 % in different agro-climatic conditions in India [3]. Application of microbial antagonists is being emerged method to

manage bacterial diseases in plants [4–7]. Several bacterial antagonists, *viz*; *Pseudomonas fluorescens*, *P. putida*, *Bacillus* spp. and *Streptomyces*, are used to control this wilt disease in tomatoes. Among them are different species of *Bacillus* like *Bacillus amyloliquefaciens*, *B. subtilis*, *B. pumilus*, *B. coagulans*, *B. cereus*, *B. licheniformis*, and *B. vallismortis* were applied for effective control of the disease [7–10]. The *Bacillus* spp. have more advantages over other genera of bacterial antagonists, like *Pseudomonas*, *Pantoea* and *Actinomycetes*, since they have resistant against desiccation and also have better survivability at higher temperatures due to possessing endospores in the cell. *Bacillus* spp. have the ability to promote plant growth slightly under adverse conditions [11–14].

Bacillus spp. suppress the growth of plant pathogens and reduce disease incidence in plants due to the production of several antibiotics, lysis of cells, a better competitor for resources, plant growth ability and also cause induced resistance in plants against the diseases. *Bacillus* species produce different kinds of antibiotics *viz*., iturin, bacillomycin, surfactin, polyketide and bacteriocins [15], lentibiotics [16]), cyclic lipopeptides [17]), polyketide mcrolectine [18,19]), phospholipid [20]), amino sugar [21]), and isocosmarin [22] to suppress various bacterial pathogens which cause diseases to the plants. Strains of *B. amyloliquefaciens* produce polyketide antibiotics *viz*; difficidin, bacillaene and macrolactin, which have antibacterial properties. Difficidin produced by *B. subtilis* and *B. amyloliquefaciens* is a highly unsaturated 22-membered macrocyclic polyene lactone phosphate ester and inhibits protein biosynthesis [23]. It has the ability to stimulate plant growth and suppress plant pathogenic organisms. The polyketide synthase *dfn* gene sequence exhibit reasonable colinearity with the polyketide structure to deduce a biosynthetic module [24]. Bacillaene gene cluster is assigned to the synthesis of the bacillaene polyketide synthase [19]. Apart from bacillaene and difficidin, a third polyketides with a macrolide-like structure, macrolactin which is produced by *B. amyloliquefacins* FZB42 contains three separate diene structure elements in a 24-membered lactrone ring [25]. It is found effective against bacterial pathogens [5]. Polyketides synthase difficidin, bacillaene [26], and macrolactin [27]) are found in the genome of *B. amyloliquefaciens* FZB42 and these gene clusters involved in the polyketides synthesis together spanning nearly 200 kb [28]. These polyketide antibiotics have a very good potential to utilize in plant disease management. There is a need to develop a technique to quickly screen potential polyketide antibiotics-producing strains of bacteria to apply for plant disease management, particularly bacterial plant diseases. In addition to this, it is also required to study the role of individual polyketide antibiotics in plant disease control, particularly the mechanism of suppression of bacterial pathogens.

The present investigations were undertaken to develop multiplex PCR-based marker for the detection of polyketides-producing strains of *Bacillus* spp. from soil samples and also study the role of polyketides antibiotics in the mechanism of antibiosis of *Bacillus amyloliquefaciens* in the suppression of bacterial wilt disease of tomato caused by *R. pseudosolanacearum*.

## 2. Materials and Methods

### 2.1. Collection of Soil Sample and Bacterial Culture

Tomato rhizospheric soil samples were collected from five states such as Uttarakhand, Meghalaya, Jharkhand, West Bengal and Odisha and one union territory Jammu & Kashmir, of India. The plants were carefully removed from the soil and the whole plants with adherent soil were kept in plastic bags for isolation of soil DNA. Bacterial cultures *viz*., *Ralstonia pseudosolanacearum* UTT-25, *Bacillus amyloliquefaciens* DSBA-11, *B. subtilis* DTBS-5, *B. cereus* JHTBS-7 and *B. licheniformis* DTBL-6 were obtained from Division of Plant Pathology, ICAR-Indian Agricultural Research Institute, New Delhi. *B. pumilus* MTCC-7092 was procured from MTCC, Institute of Microbial Technology, Chandigarh, India.

### 2.2. Antagonistic Ability of Bacillus spp. against R. pseudosolanacearum under In Vitro Conditions

The comparative antagonistic ability of *Bacillus* species *viz*., *B. amyloliquefaciens* DSBA-11, *B. cereus* JHTBS-7, *B. pumilus* MTCC-7092, *B. subtilis* DTBS-5 and *B. licheniformis* DTBL-6

were studied against bacterial wilt pathogen *R. pseudosolanacearum* using dual culture technique as described by Singh et al. [6] under in vitro conditions. The bacteria were grown in nutrient sucrose broth medium at 28 ± 1 °C for 48 h and maintained bacterial inoculums of 0.1 OD at 600 nm by spectrophotometer (UV-VIS Spectrophotometer, Hitachi, U-2900). A 100 μL culture of *R. pseudosolanacearum* was spread onto the Petri plates containing nutrient sucrose agar medium to make a lawn of the bacteria on the medium with three replications. Three wells of 5 mm diameter in each Petri plate were made with the help of a sterilized cork borer and poured 40 μL of 48 h old culture of *Bacillus* species separately containing 0.1 OD (at 600 nm) of bacterial populations into each well. The inoculated plates were kept at 28 ± 1 °C for 48 h to form an inhibition zone. The inhibition zone formed by antagonistic bacteria was measured in diameter and converted into an inhibition zone area by using the formula $\pi r^2$.

### 2.3. DNA Isolation from Bacteria

Soil samples were collected from the rhizosphere of tomatoes from Jharkhand, Odisha, West Bengal, Jammu and Kashmir, Meghalaya and Uttarakhand. The DNA of the soil was extracted by using PowerSoil DNA isolation Kits (MO BIO Laboratories, Inc.), Carlsbad, CA, USA. 0.25 g of soil sample was taken in PowerBead tubes and gently mixed, and 60 μL of solution C1 was added in each tube and vortex briefly as instruction described by the manufacturer. The extracted DNA was quantified by nanodrop for further use in PCR for the detection of bacteria-producing polyketides antibiotics. The genomic DNA of bacteria was extracted by the method described by Murry and Thompson (1980) [29].

### 2.4. Primer Designing

The three sets of primers were designed for polyketide antibiotic synthase genes *viz;* macrolactin, (mln-F and mln-R) difficidin (dfn-F and dfn-R) and bacillaene (bae-F and bae-R) from the full genome sequence of *Bacillus amyloliquefaciens* FZB-42 (Accession No. AJ634062.2) obtained from NCBI database by using Primer 3 program (www.frodo.wi.nit.edu (accessed on 12 October 2015) having product size of 792 bp, 705 bp and 616 bp respectively (Table 1). The specificity of these primers was checked by *in-silico* analysis using the website (www.insilico.ehu.es; accessed on 27 October 2015). The developed primers were validated for their universality cross *B. amyloliquefaciens* and other related groups of bacteria by primer blasting on the NCBI website (www.ncbi.nlm.nih.gov;accessed on 3 November 2015).

**Table 1.** List of polyketides primer sets used for multiplex PCR under this study.

| Primer. | Sequence (5′-3′) | Target Gene | Annealing Temp | Product Size |
|---|---|---|---|---|
| Mln-F | CGGTGATCATGAGCGCTTTG | Macrolactin | 50 °C | 792 bp |
| Mln-R | TCGGTCTGCTTTCTCAACCC | | | |
| Dfn-F | GGAAATGCCTTTAATGACC | Difficidin | 50 °C | 705 bp |
| Dfn-R | GGAGCTGAATCAATTGAAGC | | | |
| Bae-F | GTCTTACCTCGATTGCTGTG | Bacillaene | 52 °C | 616 bp |
| Bae-R | CATAGGTCACGATATCCACC | | | |

### 2.5. Standardization of Protocol for Multiplex-PCR

A PCR protocol was standardized for the simultaneous detection of polyketide antibiotic *viz;* difficidin, macrolactin and bacillaene-producing strains of *Bacillus* spp. were used to amplify from obtained template in a reaction mixture of 5 X taq buffers, 1.5 U taq polymerase (Promega), 10 mM dNTPs, 25 mM MgCl$_2$, 1 μL of each forward and reverse primers (10 μM),1 μL of DNA (100 ng) and sterile doubled distilled water in a final volume of 25 μL. The thermocycling conditions were slightly modified and consisted of one initial denaturation step of 95 °C for 5 min, followed by 32 amplification cycles of 95 °C for 30 s, 52 °C for 45 s and 72 °C for 1 min and a final extension step of 72 °C for 8 min using a BIO-RAD C1000 thermocycler. The PCR products were resolved by using a 1.5% agarose gel stained with ethidium bromide and photographed using the gel documentation system

(BIO-RAD, GEL DOCTM XRþ). The detection threshold of multiplex- PCR primer sets of polyketide genes was determined by diluting up to $10^{-3}$ of genomic DNA (100 ng) of *B. amyloliquefaciens* DSBA-11. 1.0 μL of aliquots was used as DNA template. The specificity of these primer sets was tested by using DNA templates of *B. subtilis* DTBS-5, *B. licheniformis* DTBL-6, *B. pumilus* MTCC-7092, *B. cereus* JHTBS-7, *Pseudomonas fluorescens* DTPF-3, *R. pseudosolanacearum* UTT-25, and *Xanthomonas campestris* pv. *campestris* along with *B. amyloliquefaciens* DSBA-11 for multiplex PCR as described earlier.

### 2.6. Screening of Polyketides Antibiotics Producing Strains of Bacillus spp.

Polyketide antibiotics-producing strains of *Bacillus* spp. such as *B. amyloliquefaciens* KCBA-1, KCBA-2JHBA-1, JHBA-2, MPBA-1, MPBA-2, UKTBA-1, UKTBA-2, UKTBA-3, DSBA-11 and DSBA-12, *B. subtilis* (UTTBS-1, UTTBS-2 and UTTBS-3, DTBS-4, DTBS-5, MTCC-7258), *B. cereus* JHTBS-7, *B. pumilus* MTCC-7092 and *B. licheniformis* were screened through multiplex PCR. The reaction was performed in a final volume of 25 μL amplification reaction mixture, and visualization was done as described previously.

### 2.7. Detection of Polyketides Antibiotics Producing Strains from Soil

For the detection of polyketides, antibiotics genes from extracted soil DNA was used for 25 μL of the reaction mixture of 5X taq buffers, 1.5 U taq polymerase (Promega), 10 mM dNTPs, 25 mM MgCl$_2$, 1 μL of each forward and reverse primers (10 μM) 1 μL of DNA (100 ng) and sterile doubled distilled water in a final volume of 25 μL. The thermocycling conditions were slightly modified and consisted of initial one denaturation step of 95 °C for 5 min, followed by 32 amplification cycles of 95 °C for 30 s, 52 °C for 0.45 s and 72 °C for 1 min and a final extension step of 72 °C for 8 min using a BIO-RAD C1000 thermocycler. The PCR products were resolved by using a 1.5% agarose gel stained with ethidium bromide and photographed using the gel documentation system (BIO-RAD, GEL DOCTM XRþ).

### 2.8. Molecular Characteization of Polyketides Genes in B. amyloliquefaciens DSBA-11

A study on the role of bacillaene, macrolactin and difficidin polyketides synthase genes produced by *B. amyloliquefaciens* (DSBA-11) was done for suppression of growth of *R. pseudosolanacearum*. Cloning was done by using T & A cloning vector (2.7 kb) with a competent cell of *E. coli* strain DH5α by following standard procedure as described by Sambrook et al. [30]. PCR products of these genes were purified by using RBC PCR purified kit. Transformation from ligation reaction, 10 μL of the reaction mixture was added to pre-aliquot competent cells (50 μL) and the tubes 'contents were gently mixed and placed on ice for 30 min. The cells were heat shocked in a dry water bath at 42 °C for 2 min. The tubes were transferred immediately on ice for 2 min. 1.0 mL of LB (without antibiotics) was added and the tubes were incubated for 1.5–2.0 h at 37 °C in the shaker (200 rpm). The pellets were resuspended in a suspension 100 μL of fresh LB broth and the suspension was spread at 1X gal transformation plate with a selective medium 100 mL of Luria agar medium containing 100 μL of 50 mg/mL ampicillin, 20 μL of IPTG (200 μg/mL) and 200 μL of X-gal (20 μg/mL). The plates were incubated at 37 °C for overnight and blue and white colonies appeared in the plate, which was confirmed by colony PCR with their respective primers of each polyketide antibiotic gene.

### 2.9. Bio-Efficacy of Cloned Product under In Vitro Conditions

The dual culture method was used to study the comparative antagonistic ability of cloned product of polyketide synthase antibiotic genes, including macrolactin, bacillaene and difficidin (white colonies), non-cloned (blue colonies) along with parent strain *B. amyloliquefaciens* DSBA-11 against *R. pseudosolanacearum* UTT-25 under in vitro conditions. *R. pseudosolanacearum* was grown in a CPG broth medium for 48 h at 28 ± 1 °C [6] and maintained bacterial population (0.1 OD at 600 nm). 100 μL of bacterial culture was spread onto the Petri plates containing nutrient agar medium to make a lawn of bacteria.

0.5 cm diameter of three wells in each Petri plate was made with a sterilized cork borer and 40 µL of 48 h old culture of (cloned) products of these genes (non-cloned) product and DSBA-11 grown in the Luria broth contained a population of bacteria 0.1 OD was poured into each well separately. The plates were incubated at 28 ± 1 °C for 48 h and the inhibition zone was recorded. The value of the inhibition zone was converted into area of inhibition zone as described by Singh et al. [6].

### 2.10. Bacterial Viability Test

The *R. pseudosolanacearum* culture was treated with 10 and 50 µg/mL of cloned macrolactin, bacillaene, difficidin, *B. amyloliquefaciens* DSBA-11 and a mixture of these three polyketide antibiotic synthase genes to see the growth variation at different intervals 24 h, 48 h, 72 h, and 96 h after treatments and OD at 600 nm was taken by using UV-VIS Spectrophotometer, Hitachi (U-2900).

### 2.11. Mechanisms of Suppression of Bacterial Wilt Disease of Tomato

The cloned product of macrolactin, difficidin and bacillaene genes, *B. amyloliquefaciens* (DSBA-11) and non cloned product were tested against *R. pseudosolanacearum* under controlled conditions at National Phytotron Facility, IARI New Delhi. 21 days old tomato cv. Pusa Ruby seedlings were transplanted in the 6′ plastic pots having autoclaved soil mixture of peat moss, vermiculite and sand in the ratio of 2:1:1 at 25–30 °C. *R. pseudosolanacearum* UTT-25, *B. amyloliquefaciens* (DSBA-11), cloned and non-cloned colonies of bacteria were scraped from the plates containing LA (Luria Agar) medium and suspended in sterile distilled water and maintained the population of bacteria (0.1 OD at 600 nm). A 40 mL of 48 h old culture of *R. pseudosolanacearum* UTT-25 was inoculated at the root zone of each plant after 5 days of transplanting. Subsequently, 25 mL of 48 h old culture of antagonistic *B. amyloliquefaciens* (DSBA-11), cloned and non-cloned product of three polyketide antibiotic genes was inoculated at the root zone of each plant. The plants inoculated with *R. pseudosolanacearum* and un-inoculated plants were also maintained as the positive and negative control. Wilt disease intensity was recorded at 3 days intervals, up to 30 days. The wilt disease intensity and rating were calculated as per described by Schaad et al. [31], and biological control efficacy (BCE) of antagonistic bacteria was calculated as per Guo et al. [32], and Singh et al. [7].

### 2.11.1. Purification of Cloned Product of Polyketides Antibiotic Synthase Gene

To purify the cloned polyketide antibiotic genes of *B. amyloliquefaciens* DSBA-11, the colony was grown in an LB medium and incubated for 48 h. After that the liquid culture was eluted in 100% methanol by using GC/MS analysis on a Focus GC/MS (Thermo) equipped and DB-5 capillary column (30 m × 0.25 mm i.d., film thickness 0.25 µm). Chromatographic conditions were as follows: helium as carrier gas at a flow rate of 1 mL/min; injection volume was 1.0 µL (1000 ppm in acetone); injector temperature was 240 °C, respectively. The column temperature was held at 60 °C for 5 min., and programmed at 3 °C/min to 220 °C and held for 10 min with split mode (1:20). The MS transfer line and source temperatures were 240 °C and 220 °C. The analysis was carried out in EI mode at 70 eV with the mass range of 30–450 a.m.u at 1 scan/s. The individual cloned polyketides antibiotic genes were purified and the extracts were used for the bio-efficacy test against *R. pseudosolanacearum*.

### 2.11.2. TEM Studies

Transmission electron microscopy (TEM) analysis was used to determine the effect of the eluted cloned product of polyketide genes macrolactin; difficidin, and bacillaene on *Ralstonia pseudosolanacearum* UTT-25 cells at the ultra structural levels. *R. pseudosolanacearum* UTT-25 was treated with 50 µg/mL of these antibiotics along with parent *B. amyloliquefaciens* DSBA-11 and allowed them to grow for 14 h at 28 ± 1 °C. One drop of the sample was embedded on a copper grid having 400-mess for 1–2 min and thoroughly washed with

sterile distilled water and then negative stain with 2% uranyl acetate and dried it. The grid was visualized under transmission electron microscope (TEM) model no-Jeol-1011.

### 2.11.3. Statistical Analysis

The data was analyzed using Fisher's least significant differences (LSD) to determine the significant differences between treatments at $p < 0.05$ level.

## 3. Results

### 3.1. In Vitro Studyon Antagonistic Ability of Bacillus spp. against R. pseudosolanacearum

Five *Bacillus* species *viz.*, *Bacillus amyloliquefaciens* DSBA-11, *B. cereus* JHTBS-7, *B. pumilus* MTCC-7092, *B. subtilis* DTBS-5 and *B. licheniformis* DTBL-6 were used against *R. pseudosolanacearum* UTT-25 under in vitro conditions. A significant variation (CD value (5%): 0.355, CV: 7.14%) was recorded among the species of *Bacillus* to form an inhibition area against *R. pseudosolanacearum*. Maximum inhibition area against *R. pseudosolanacearum* was recorded in the treatment of *B. amyloliquefaciens* DSBA-11 (3.30 cm$^2$) followed by *B. subtilis* DTBS-5 and *B. licheniformis* DTBL-6 after 48 h of incubation at 28 ± 1 °C (Figure 1).

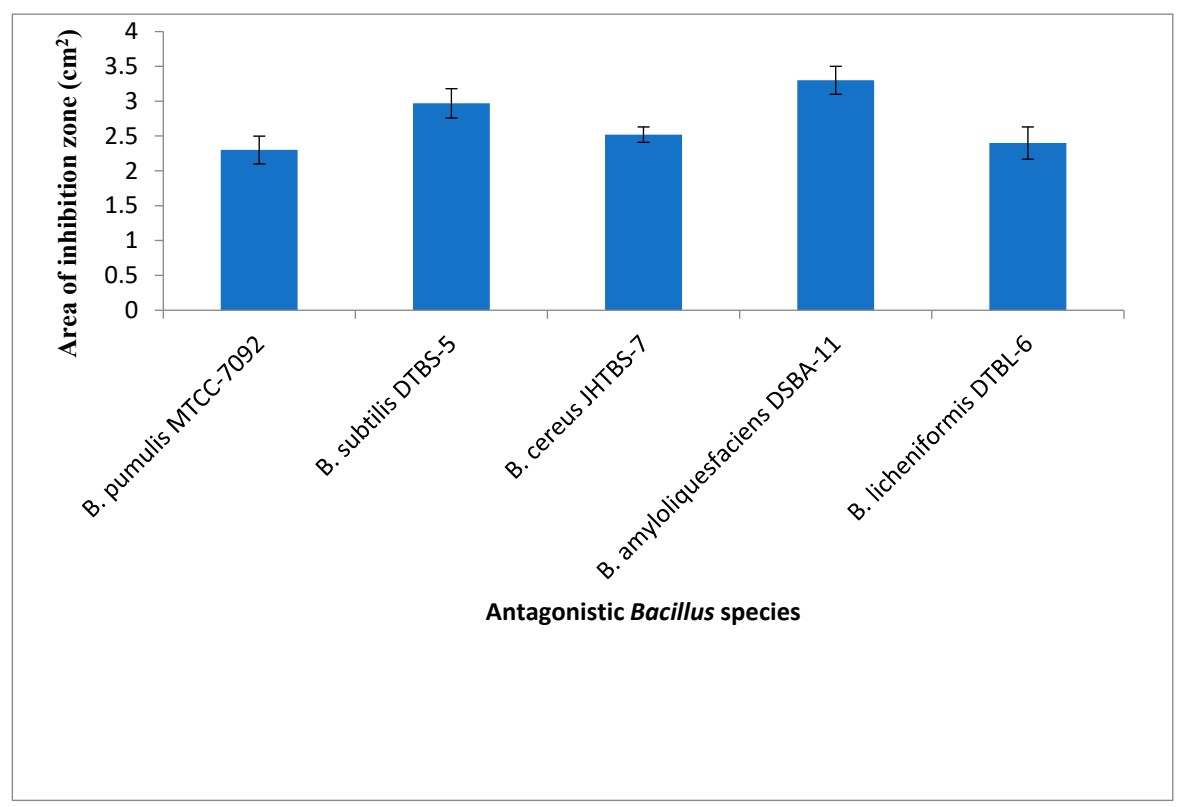

**Figure 1.** The antagonistic ability of *Bacillus* spp. against R. *pseudosolanacearum* under in vitro conditions.

### 3.2. Primer Specificity and Sensitivity

The primer sets *viz.* Mln-F &Mln-R, Dfn-F &Dfn-R and Bae-F & Dfn-R-based on the nucleotide sequence of macrolactin, difficidin and bacillaene genes of *B. amyloliquefaciens* amplified at 792 bp, 705 bp, and 616 bp respectively (Table 1). To check the specificity these primer sets, three banding pattern in *B. amyloliquefaciens* strains were obtained at 792 bp, 705 bp and 616 bp by amplifying macrolactin, difficidin, and bacillaene polyketide synthase genes respectively. *B. subtilis* strains DTBS-4 & DTBS-5 amplified macrolectin, and difficidin synthase genes at 792 bp and 705 bp, respectively. However, these primer pairs of polyketide antibiotic synthase genes did not give amplification with other *Bacillus* spp. like *B. pumilus* MTCC 7092, *B. licheniformis* DTBL-6 and *B. cereus* JHTBS-7, *P. fluorescens* DTPF-3, *R. pseudosolanacearum* UTT-25 (Figure 2). Thus, these three sets of primers-based

on macrolactin, difficidin and baciallene genes were specific for the detection of *B. amyloliquefaciens*. The diluted genomic DNA of *B. amyloliquefaciens* was used for the sensitivity of polyketide antibiotic synthase-based primers. These polyketide antibiotic synthase genes-based primers were able to detect low to 0.1 ng DNA concentrations of *B. amyloliquefaciens* (Figure 3).

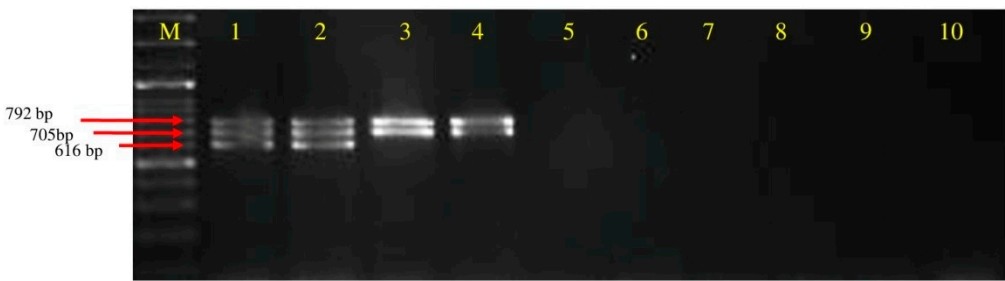

**Figure 2.** Simultaneous amplification of three polyketide antibiotic synthase genes i.e., difficidin, macrolactin and bacillaene of *Bacillus* at 705 bp, 792 bp and 616 bp using multiplex—PCR for primer specificity, respectively. Lane 1; M. 100 bp DNA Ladder: Lane 1: *B. amyloliquefaciens* DSBA-11; 2: *B. amyloliquefaciens* DSBA-12; 3: *B. subtilis* (DTBS-4); 4 *B. subtilis* DTBS-5; 5: *B. pumilus* MTCC-7092; 6: *B. licheniformis* DTBL-6; 7: *B. cereus* (JHTBS-7); 8: *Pseudomonas fluorescens* DTPF-3; 9: *R. pseudosolanacearum* UTT-25; 10: -ve control.

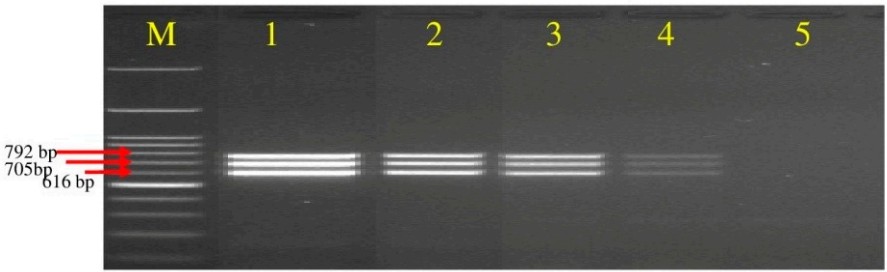

**Figure 3.** Sensitivity of the primers designed based on three polyketide antibiotic synthase genes i.e., difficidin, macrolactin and bacillaene of *Bacillus* amplified at 705 bp, 792 bp and 616 bp using multiplex–PCR, respectively. Lane M: 100 bp DNA Ladder: Lane 1: 100 ng/μL DNA of *B. amyloliquefaciens* (DSBAS-11); 2: 10 ng/μL; 3: 1.0 ng/μL; 4: 0.1 ng/μL 5: -ve control.

*3.3. Screening of Polyketides Antibiotics Producing Strains of Bacillus spp. through Multiplex PCR*

A combination of two bands of macrolactin + bacillaene, bacillaene + difficidin and difficidin + macrolactin was also observed. However, the amplification of such type of banding pattern was lower. The positive control of *B. amyloliquefaciens* DSBA-11 and DSBA-12 showed amplification with these three primers sets and negative control did not produce any band. Out of twenty-three isolates of *Bacillus* spp., 11 isolates of *B. amyloliquefaciens*, 6 isolates of *B. subtilis*, three strains of *B. cereus*, one strain each of *B. pumilus* and *B. licheniformis* were screened using multiplex PCR. All the 11 isolates of *B. amyloliquefaciens* from different hosts namely KCBA-1, KCBA-2, JHBA-1, JHBA-2, MPBA-1, MPBA-2, UKBA-1, UKBA-2, UKBA-3, DSBA-11 andDSBA-12 were amplified at 792 bp, 702 bp and 616 bp (Table 2) and produced all three antibiotics; while *B. subtilis* isolates including UTTBS-1, UTTBS-2, UTTBS-3, DTBS-4, DTBS-5, *B. subtilis* MTCC 7258, were amplified two set of primers based on macrolactin and difficidin genes. Other species of *Bacillus* like *B. pumilus*, MTCC-7092, *B. cereus* JHTBS-7 and *B. licheniformis* DTBL-6, did not show amplification of these genes (Figure 4; Table 3).

**Table 2.** List of soil DNA sample used for the detection of polyketides genes by multiplex-PCR.

| Soil Sample | Place | No of Soil Samples Tested | All Three Genes | Polyketides Genes | | |
|---|---|---|---|---|---|---|
| | | | | Bae + Dfn | Dfn + Mln | Mln + Bae |
| 1 | Jharkhand | 16 | 62.5 (10) | 37.5 (6) | 0.0 | 0.0 |
| 2 | Odisha | 12 | 58.3 (7) | 16.0 (2) | 8.4 (1) | 16.8 (2) |
| 3 | West Bengal | 15 | 73.3 (11) | 0.0 | 13.3 (2) | 13.3 (2) |
| 4 | Jammu & Kashmir | 7 | 66.7 (4) | 16.7 (1) | 16.7 (1) | 16.7 (1) |
| 5 | Meghalaya | 5 | 60.0 (3) | 0.0 | 20.0 (1) | 20.0 (1) |
| 6 | Uttarakhand | 20 | 70.0 (14) | 5.0 (1) | 10.0 (2) | 15.0 (3) |
| 7 | *B. amyloliquefaciens* DSBA-11 and DSBA-12 (positive) | – | + | 0.0 | 0.0 | 0.0 |
| | Total | 75 | | | | |

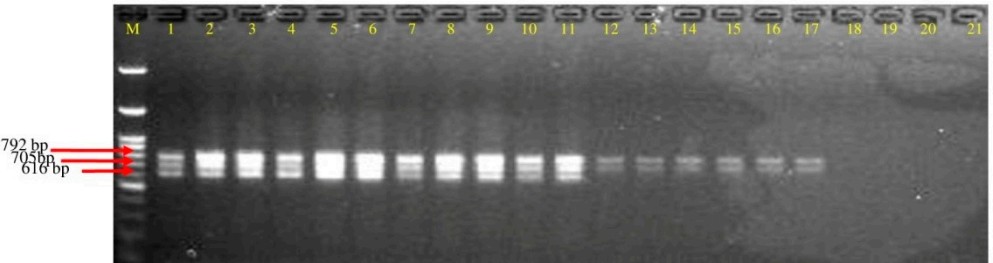

**Figure 4.** Detection *Bacillus* species using multiplex- PCR based on polyketides genes i.e., difficidin, macrolactin and bacillaene of *Bacillus* amplified at 705 bp, 792 bp and 616 bp respectively. Lane M. 100 bp DNA Ladder: lane 1: KCBA-1, 2: KCBA-2, 3: JHBA-1, 4: JHBA-2, 5: MPBA-1, 6: MPBA-2, 7: UKBA-1, 8: UKBA-2, 9: UKBA-3,10: *B. amyloliquefaciens* DSBA-11, 11: *B. amyloliquefaciens* DSBA-11 12: UTTBS-1, 13: UTTBS-2, 14: UTTBS-3, 15: DTBS-4, 16: DTBS-5, 17: *B. subtilis* MTCC 7258, 18: *B. cereus* JHTBS-7, 19: *B. pumilus* MTCC-7092, 20: *B. licheniformis* DTBL -6, 21: Negative control.

**Table 3.** Screening of polyketide antibiotics producing strains of *Bacillus* spp. isolated from rhizospheric soil of solanaceous crops.

| Polyketides Producing Strain of *Bacillus* spp. | No of Isolates Tested | Host | Place | Polyketides Genes | | |
|---|---|---|---|---|---|---|
| | | | | Bae | Dfn | Mln |
| *B. amyloliquefaciens* | KCBA-1, KCBA-2 | Chilli rhizosphere | Karnataka | + | + | + |
| | JHBA-1, JHBA-2 | Brinjal rhizosphere | Jharkhand | + | + | + |
| | MPBA-1, MPBA-2 | Potato rhizosphere | Meghalaya | + | + | + |
| | UKTBA-1, UKTBA-2, UKTBA-3 | Tomato rhizosphere | Uttarakhand | + | + | + |
| | DSBA-11, DSBA-12 | | Delhi | + | + | + |
| *B. subtilis* | UTTBS-1, UTTBS-2 UTTBS-3, DTBS-4, DTBS-5 | Tomato rhizosphere | Uttarakhand & Delhi | – | + | + |
| *B. subtilis* | MTCC 7258 | – | MTCC Chandigarh | – | + | + |
| *B. cereus* | JHTBS-7 | Brinjal rhizosphere | Jharkhand | – | – | – |
| *B. pumilus* | MTCC-7092 | – | MTCC Chandigarh | – | – | – |
| *B. licheniformis* | | – | Bacteriology, Div. of plant pathology | – | – | – |

### 3.4. Detection of Polyketides Gene from Soil DNA

Eighty-five samples were collected from six states of India to detect polyketide-producing strains of *Bacillus* spp. Maximum 73.3% of soil samples collected from West Bengal showed potential to amplify with all these primers followed by Uttarakhand (70.0%) and Jammu and Kashmir (66.7%) (Table 3; Figure 5).

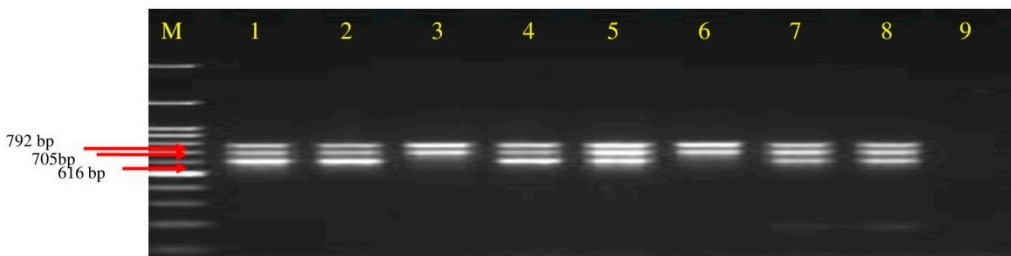

**Figure 5.** Detection of *Bacillus* species from soil samples collected from different states of India by using multiplex- PCR based on polyketide antibiotic synthase genes i.e., difficidin, macrolactin and bacillaene of *Bacillus* amplified at 705 bp, 792 bp and 616 bp respectively. Lane; M. 100 bp DNA Ladder: lane 1–2: *B. amyloliquefaciens* DSBA-11 and DSBA-12; 3: West Bengal; 4: Jharkhand; 5: Odisha; 6: Meghalaya; 7: Uttarakhand; 8: J&K; 9: -ve control.

### 3.5. Bio-Efficacy of Cloned Product of Polyketides Antibiotics Producing Gene under In Vitro Conditions

The polyketides antibiotic synthase gene difficidin, macrolactin and bacillaene of *B. amyloliquefaciens* was cloned cloning and confirmed by colony PCR by using their respective genes with the standard PCR protocol with their respective primers to amplified at 792 bp 705 bp and 616 bp respectively. The cloned product of these genes suppressed the growth of *R. pseudosolanacearum* significantly under in vitro conditions. Maximum growth inhibition of *R. pseudosolanacearum* was found in parent *B. amyloliquefaciens* DSBA-11 3.30 cm$^2$ of inhibition zone of *R. pseudosolanacearum*. However, the cloned product of bacillaene, macrolactin and difficidin formed 1.9, 1.9 and 1.7 cm$^2$ of inhibition zone area under in vitro conditions, respectively (Figure 6B–D). However, these genes alone made less inhibition zone as compared to their parent. A significant variation in the growth of *R. pseudosolanacearum* was recorded in different antibiotic treatments. The growth of bacteria was increased by increasing the duration of incubation in all the treatments, including control significantly (Table 4).

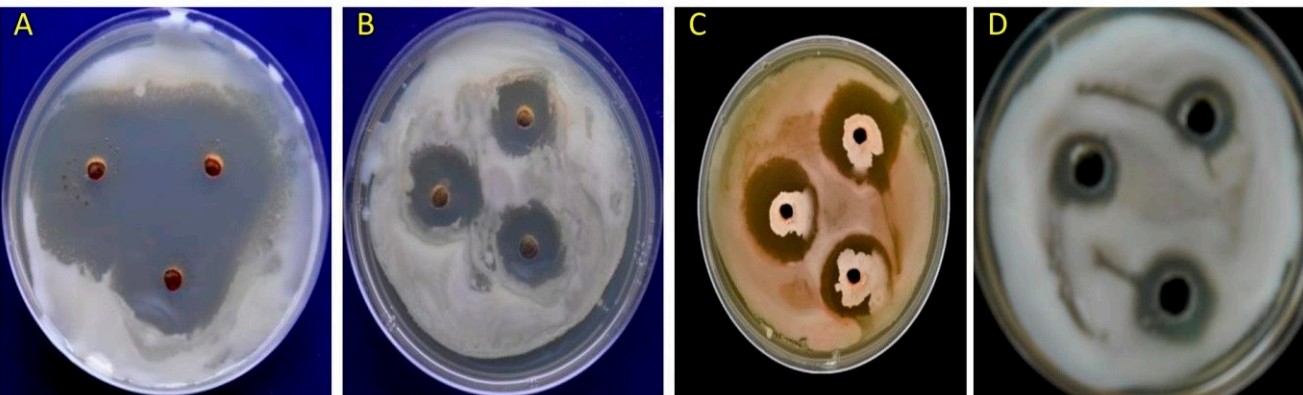

**Figure 6.** Inhibition zone formed by clone product of polyketide antibiotic synthase genes of *Bacillus amyloliquefaciens* DSBA-11 against *R. pseudosolanacearum*. (**A**) *B. amyloliquefaciens* DSBA-11 (Wild strain) (**B**) clone of difficidin gene (**C**) clone of macrolactin gene (**D**) bacillaene gene.

**Table 4.** Formation of inhibition zone and reduction of bacterial wilt disease intensity by-product of polyketide antibiotics synthase gene cloned and parent strain of *B. amyloliquefaciens* DSBA-11.

| Treatments | Inhibition Zone (Area in cm²) | Wilt Disease Intensity (%) | Biocontrol Efficacy (%) |
|---|---|---|---|
| *B. amyloliquefaciens* DSBA-11 (Parent) | 3.3 [a] | 29.3 [f] | 57.72 |
| Cloned product ofbacillaene (bae) | 1.9 [c] | 50.7 [c] | 26.83 |
| Cloned product ofmacrolactin (mln) | 1.9 [c] | 46 [d] | 33.62 |
| Cloned product of difficidin (dfn) | 1.7 [c] | 39.3 [e] | 43.23 |
| *R. solanacearum* UTT-25 | 0.0 [b] | 69.3 [a] | - |
| Non-clonedproduct | 0.0 [b] | 66.7 [b] | - |

Dissimilar alphabetical letters in each column have significant statistical differences ($p \leq 0.05$) at the level using Tuckeys multiple range test (TMRT).

*3.6. Morphological and Ultrastructural Changes of R. pseudosolanacearum Cells in the Presence of Polyketide Antibiotics*

Visualization of the cellular damage caused to *R. pseudosolanacearum* UTT-25 at the ultrastructural level by difficidin, macrolactin and bacillaene was undertaken through TEM analysis (Figure 7). In this study, untreated control cell of *R. pseudosolanacearum* UTT-25 appeared intact, typically rod-shaped with smooth exterior (Figure 7A). Upon exposure to difficidin, macrolactin and bacillaene antibiotics, the cells of *R. pseudosolanacearum* became loose and diffused with their normal shape and even ruptured (Figure 7B–F). After treatment of *R. pseudosolanacearum* UTT-25 with these polyketide antibiotics, the lysis of the bacterial cells was clearly visible and resulted in plasmolysis and efflux of intracellular components.

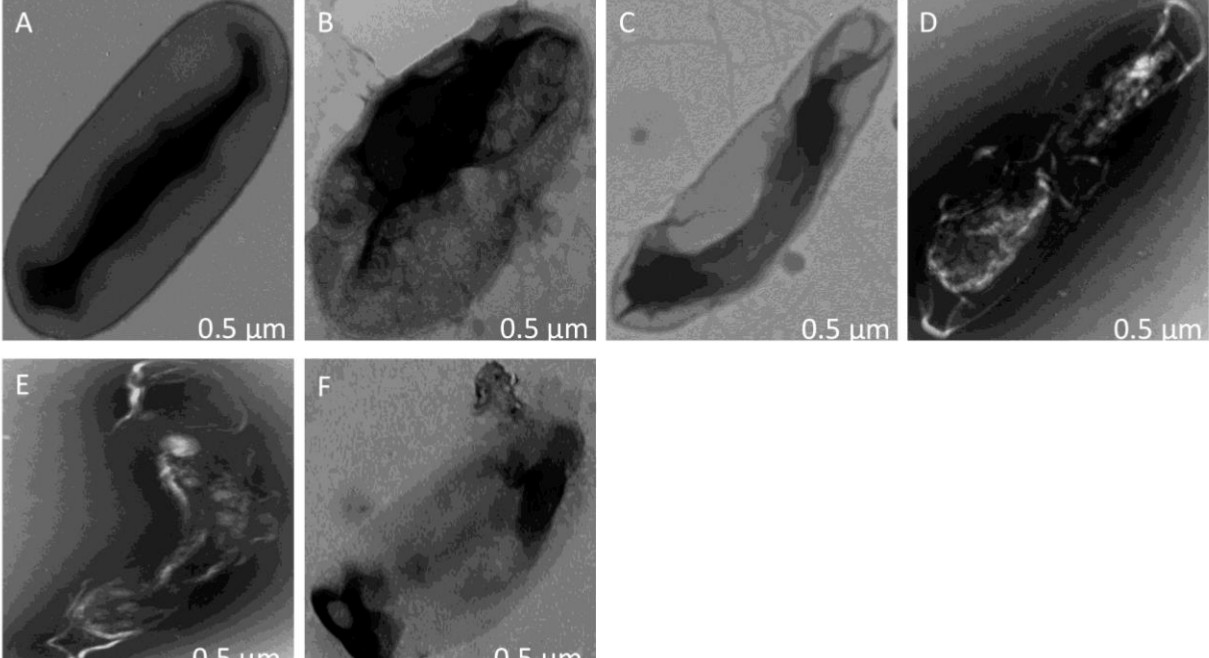

**Figure 7.** Ultrastructural effect of 50 μg/mL concentration of difficidin, bacillaene and macro-lactin, of *B. amyloliquefaciens* DSBA-11 and mixture of difficidin, bacillyene, macrolactin on *R. pseudosolanacearum* cells after 14 h determined by TEM. (**A**) an untreated *R. solanacearum cell* (**B**) *R. pseudosolanacearum* treated with macrolactin (**C**) *R. pseudosolanacearum* treated with difficidin (**D**) *R. pseudosolanacearum* treated with bacillaene (**E**) *R. pseudosolanacearum* treated with Parent *B. amyloliquefaciens* DSBA-11 (**F**) *R. pseudosolanacearum* treated with a mixture of bacillaene, macrolactin and difficidin. Bars: 0.5 μm.

### 3.7. Effect of Polyketide Antibiotic-Producing B. amyloliquefaciens for Control of Bacterial Wilt

The mechanism of suppression of wilt disease in tomato, the cloned product of bacillaene, difficidin and macrolactin genes along with their parent *B. amyloliquefaciens* DSBA-11 and the non-cloned product was inoculated to the plant. A minimum 29.3% of wilt intensity in tomato plants was recorded in treated with *B. amyloliquefaciens* DSBA-11 under glasshouse conditions. The treatment of cloned product of difficidin, macrolactin and bacillaene synthase genes of *B. amyloliquefaciens* DSBA-11 significantly suppressed 39.3, 46.0, and 50.7% of wilt intensity in tomato cv Pusa Ruby after 30 days of *R. pseudosolanacearum* UTT-25 inoculation respectively (Table 4). However, the wilt intensity was found to be lower in cloned products of polyketide antibiotic synthase genes than control (69.3%). The non-cloned colony of bacteria slightly showed low wilt intensity in tomato as compared to control but lower than the cloned product.

### 3.8. Effect of Macrolactin, Difficidin and Bacilysin on Viability of R. pseudosolanacearum Cells

Growth of *R. pseudosolanacearum* UTT-25 treated with crude ethyl acetate extract of polyketide of these antibiotics genes produced by *B. amyloliquefaciens* DSBA-11 and its cloned products in broth culture was measured by using optical density measurements at 600 nm (Table 5). Minimum growth of *R. pseudosolanacearum* was found at 50 µg/mL of a mixtures of macrolactin + difficidin + bacillaene 0.063 OD (at 600 nm) after 24 h of incubation followed by *B. amyloliquefaciens* DSBA-11 (0.071 OD) and difficidin (0.071 OD) under in vitro conditions. The growth of *R. pseudosolanacearum* UTT-25 was inhibited by increasing the concentration of polyketide antibiotic products.

**Table 5.** Effect of crude ethyl acetate extracts of polyketide antibiotics of *B. amyloliquefaciens* DSBA-11 and its cloned product on the growth of *R. pseudosolanacearum* under in vitro conditions.

| Treatment | Concentration (µg/mL) | Growth of *R. pseudosolanacearum* OD Value at 600 nm | | | |
|---|---|---|---|---|---|
| | | 24 h | 48 h | 72 h | 96 h |
| Macrolactin | 10 µg/mL | 0.168 [bc] | 0.197 [c] | 0.244 [c] | 0.250 [c] |
| | 50 µg/mL | 0.075 [cd] | 0.077 [d] | 0.093 [gh] | 0.095 [f] |
| Difficidin | 10 µg/mL | 0.190 [b] | 0.275 [b] | 0.334 [b] | 0.339 [b] |
| | 50 µg/mL | 0.071 [cd] | 0.082 [d] | 0.110 [ef] | 0.116 [ef] |
| Bacillaene | 10 µg/mL | 0.181 [b] | 0.247 [b] | 0.252 [c] | 0.255 [c] |
| | 50 µg/mL | 0.077 [cd] | 0.088 [d] | 0.133 [e] | 0.136 [e] |
| Macrolactin + Difficidin + Bacillaene | 10 µg/mL | 0.074 [cd] | 0.086 [d] | 0.090 [gh] | 0.094 [f] |
| | 50 µg/mL | 0.063 [d] | 0.070 [d] | 0.076 [g] | 0.079 [f] |
| *B. amyloliquefaciens* DSBA-11 | 10 µg/mL | 0.128 [bcd] | 0.170 [c] | 0.190 [d] | 0.195 [d] |
| | 50 µg/mL | 0.071 [cd] | 0.080 [d] | 0.102 [gh] | 0.107 [ef] |
| Control | | 1.76 [a] | 2.603 [a] | 2.88 [a] | 2.98 [a] |

Means followed by the same letter within a column are not significantly different as determined by LSD test ($\alpha$ = 0.05). Data present means of the experiment within 3 replications each.

## 4. Discussion

Bacterial wilt of solanaceous crop caused by *Ralstoniapseudosolanacearum* UTT-25 is a serious problem across the world. For the control of wilt disease, we selected *Bacillus* spp. particularly polyketide antibiotic-producing strains of *Bacillus* spp. as earlier, it was reported that *Bacillus* spp. produced 200 peptide antibiotics (Lisboa et al. [33]), and approximately 8.5% genome of *B. amyloliquefaciens* FZB 42 are dedicated to secondary metabolite production (Chen et al. [5]. Selection of potential antibiotic-producing strains of *Bacillus* spp., the individual gene had been targeted to design primer for detection like iturin. (6)], and bacillaene [34]. In this study, we targeted three polyketide antibiotic synthase genes viz., macrolactin, difficidin and bacillaene for developing a multiplex-PCR protocol to detect polyketide antibiotic-producing strains of *Bacillus* spp. We developed and standardized a protocol for detecting and screening of polyketide-producing strains of *Bacillus* spp. by using multiplex—PCR, based on these genes, which is rapid, accurate and highly sensitive. To the best of our knowledge, this is the first attempt to develop a protocol for simultaneous detection of these three polyketide antibiotic-producing strains of

*Bacillus* spp. particularly *B. amyloliquefaciens.* We screened polyketide antibiotic-producing strains of *Bacillus* spp. including *B. amyloliquefaciens*, *B. subtilis*, *B. cereus*, *B. pumilus*, and *B. licheniformis* using multiplex- PCR and among them, the Isolates of *B. amyloliquefaciens* contained macrolactin, bacillaene and difficidin polyketide synthase antibiotic genes, while *B. subtilis* contained difficidin and macrolactin synthase antibiotic genes. In contrast to our results, Butchner et al. [34] reported that approximately 80 kb pksX gene cluster in *B. subtilis* encode an unusual hybrid polyketide that has been linked to the production of the uncharacterized antibiotic bacillaene.

The most attention has been paid to polyketide antibiotics such as bacillaene, difficidin and macrolactin for their role in the inhibition of bacterial growth and inducing plant resistance against the bacterial pathogen [35] and difficidin and bacilysin [36]. Antibiotics produced by the strains of different bacterial genera, biosynthetic genes have been cloned partially and sequenced. However, *B. amyloliquefaciens* FZB 42 is a producer of three families of lipopeptides, surfactin, bacillomycin D, and fengycin, which are well-known secondary metabolites with mainly antifungal and bacterial activities. They are also produced by numerous *B. subtilis* strains. Furthermore, three giant gene clusters containing genes with homology to polyketide synthase (PKS) genes have been identified but did not assign functional roles. Polyketides belong to a large family of secondary metabolites that include many bioactive compounds with antibacterial, immune suppressive, antitumor, or other physiologically relevant bioactivities. Three functional gene clusters directing the synthesis of difficidin, bacillaene, and macrolactin have been identified in *B. amyloliquefaciens* strains as earlier reported by Chen et al. [5]. Further, polyketide mega synthesis responsible for synthesis of bacillaene, macrolactin and difficidin are encoded by these giant gene clusters that are located at different sites of the *B. amyloliquefaciens* genome [35]. Difficidin inhibits protein synthesis and possibly also damages the cell membrane [36,37]. Polyketide synthase (PKSs) gene clusters has also been identified that directed the synthesis of polyketides (PKs), e.g., macrolactin (mln), bacillaene (bae), and difficidin in *B. amyloliquefaciens* subsp. *plantarum* FZB42 [38]. The present study showed the antagonistic ability of cloned products of macrolactin, bacillaene and difficidin along with parent DSBA-11, which showed an inhibitory effect on the growth of *R. pseudosolanacearum* and also suppressed wilt disease in tomatoes. We cloned macrolactin, difficidin and bacillaene genes of *B. amyloliquefaciens* DSBA-11 and we observed that the product of these genes and parent DSBA-11 damaged the cell wall and cell membrane of *R. pseudosolanacearum*, which were visualized under TEM (Figure 4). Similar results have also been reported by Wu et al. [36]. The findings were corroborative to earlier reports on difficidin, which acts as antibacterial activity against *Xanthomonas oryzae* pv. *oryzae* and *X. oryzae* pv. *oryzicola* causing bacterial leaf blight and bacterial streak disease in rice, respectively. Although, no significant variation was found in the cloned products to inhibit the growth of *R. pseudosolanacearum* under in vitro conditions. However, difficidin was found to be the best among these tested polyketide antibioticsto suppress the bacterial wilt disease in tomatoes. Moreover, these cloned products showed less biocontrol efficacy than parent strain DSBA-11 (57.22%). A similar finding was earlier reported by Chen et al. [35] against *E. amylovora*, causing fire blight in pears and apples. They stated that difficidin is the most efficient antibacterial compound in FZB42. Our present finding reveals that the *B. amyloliquefaciens* DSBA-11 contains these three polyketide antibiotics, which are able to inhibit the growth of *R. pseudosolanacearum* under in vitro conditions by forming an inhibition zone and reduce wilt disease in tomatoes significantly. Further, structural and functional characterization of these polyketide antibiotic synthase genes is required for a detailed study.

## 5. Conclusions

The following conclusions can be drawn from the study results:

1. *Bacillus amyloliquefaciens* DSBA-11 showed the best antagonistic ability against *R. pseudosolanacearum* UTT-25 under in vitro and in vivo conditions.

2. A multiplex—PCR protocol based on polyketide synthase genes i.e.,macrolactin, difficidin and bacillaene genes, was developed to identify and detect potential antagonistic strains of *B. amyloliquefaciens* within a short period of time with high accuracy and sensitivity.

3. *Bacillus amyloliquefaciens* DSBA-11 provides a potential option to use these polyketide antibiotics as an alternative to chemical bactericides to control wilt disease of solanaceous crops.

4. There is a need to study the mechanism of polyketide antibiotic synthase genes *viz.*, macrolactin, difficidin and bacillaene using high throughput molecular techniques.

**Author Contributions:** D.S. conceived and designed the experiments. V.D. did the planning and edited the manuscript. D.K.Y. performed most of the experiments such as screening of antibiotics genes and cloning of polyketide antibiotic synthase genes. All authors have read and agreed to the published version of the manuscript.

**Funding:** The funds received from the Outreach project on *Phytophthora*, *Fusarium* and *Ralstonia* Diseases of Horticultural and Field Crops sponsored by ICAR, New Delhi for conducting various experiments.

**Acknowledgments:** The authors are grateful to the Indian Council of Agricultural Research, New Delhi, to giving financial assistance under outreach project on *Phytophthora*, *Fusarium* and *Ralstonia* Diseases of Horticultural and Field Crops sponsored by ICAR, New Delhi for conducting these experiments. The authors are also thankful to the Head, Division of Plant Pathology, Indian Agricultural Research Institute, New Delhi, for helping throughout the experimentation.

**Conflicts of Interest:** The authors declare that they have no conflict of interest.

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
