# Peer review of "Suppression of Tomato Bacterial Wilt Incited by Ralstonia pseudosolanacearum Using Polyketide Antibiotic-Producing Bacillus spp. Isolated from Rhizospheric Soil"

_agriculture, doi:10.3390/agriculture12122009_

Round 1

Reviewer 1 Report

The authors had done significant research. The Abstract need to be modified and shorten by only keeping key results. The entire Ms. need through English proofreading by native speaker or professional services since there are several grammatical and punctuation errors eg. line no.113 g should be written for gm and Power in place of poer. Fig 1 is not necessary and may be removed. Discussion need to be improved by adding recent and relevant studies.

Author Response

Reviewer 1

1. The Abstract need to be modified and shorten by only keeping key results.
                       Revised abstract.
1 The entire Ms. need through English proofreading by native speaker or professional services since there are several grammatical and punctuation.
The manuscript has been checked by all the authors carefully and corrected accordingly.
2 line no.113 g should be written for gm and Power in place of power.
          Corrected in the manuscript
3 Fig 1 is not necessary and may be removed.
                       Fig. 1 has been deleted from MS
4 Discussion needs to be improved by adding recent and relevant studies.
           Discussion has been improved by adding recent and relevant studies

Reviewer 2 Report

Dinesh et al. seek to elucidate the suppression mechanism of bacterial wilt disease suppression in tomato by Bacillus spp. isolated from rhizospheric soil, and design three primers to identify Bacillus amyloliquefaciens. The authors cloned the key genes in E. coli, and measure the inhibition ability to pathogens. They also isolated the polyketide form recombination strains, and compared the biocontrol efficacy across three products.

Overall, I feel that the topic is relevant to the readership of Agriculture. I think the authors could make a stronger argument for why this new information is relevant in discussion. As it currently reads, it seems that the results are mainly confirming what has previously been discovered in other studies, rather than highlighting what is new and relevant from their observations in the Discussion. The results of antagonistic ability of Bacillus spp. against R. pseudosolanacearum lacked statistical rigor. The manuscript was poorly written and organized, with a large number of spelling and grammatical errors.

Specific comments:

Abstract

1. Line 11-17: Bacillus amyloliquefaciens DSBA-11 showed best inhibition to R. pseudosolanacearum, and why use strain FZB42 to design primer sets? The rationale for why this species was selected is poorly defined.

2. The abbreviations of strains should be uniform, and the name of genus is the full name when it first appears. some name of strains are nonstandard or misspelled, such as B subtilis on line 20.

3. Line19-21: How many strains of B. subtilis was used, they all possess difficidin and macrolactin. And only one strain MTCC-7092 belonging to B. pumilus was used?

4. Line 25: Many type mistakes in the manuscript, such as comma after "bacillaene" should be not superscript. And replace "product" with "products".

6. Capitalization should be uniform in Keywords.

Introduction

7. Line44: Some words are too colloquial, such as “very heavy” on line 44, “very good” on line 74-75.

8. “Among them different species of Bacillus like B. amyloliquefaciens, B. subtilis, B. pumilus, B. coagulans, B. cereus, B. licheniformis, and B. vallismortis were applied for effective control of the disease [8, 9, 10, 7]. However, the Bacillus spp. have more advantages over other genera of bacterial antagonists,…”: There is no transition between the first sentence and the next sentence. The word "however" should be deleted.

9. Line 57: The word "food" is inappropriate here, and "resources" may be more appropriate.

10. Line 66: what the meaning of “co linearity”?

11. Line 68: delete “.” After “[19]”.

Materials and Methods

12. Line 113: The name kits should be checked. Line 123: replace “data base” with “database”.

13. Line 133: a comma should be added before "1 ul".

14. Line 136: annealing time "0.45 sec "is too short, "45 sec"?

15. Figure 1: Move Figure 1 to supplementary materials or delete.

16. Replace “such as” with “including” on line 184, 355.

17. Line 190: what the meaning is "old culture"? How to determine the concentration of the compound.

18. The methods of 2.10 and 2.11.3 are repeated.

19. Line 220: How to purify a gene? or you mean purify the product of polyketide? And usually “product of a gene” is protein. In here, the product may be polyketide, change the formulation of “product of gene”.

Results

20. Line 248: The results of antagonistic ability of Bacillus spp. against R. pseudosolanacearum lacked statistical rigor.

21. Figure 2: The brackets in the ordinate are incomplete. Statistical analysis should be done between different species.

22. Line 255-260: The genes macrolactin, and difficidin were also detected in DTBS-4 and DTBS-5. it is inappropriate to say the primers were specific for B. amyloliquefaciens.

23. Line 262: change “up to” to “low to”?

24. Figure 3 and 4: It is strange that in Figure 3, the DNA ladder is 100bp, while in Figure 4, it is 1kb. It seems that they are the same. Why there are 792, 705 and 616 bp bands in DNA marker. Marking the size of each stripe will make the results clearer.

25. Line 299: Delete “cloning”.

26. Table 4: The error value of inhibition should be added.

27. Figure 7: A control with no-clone is missing. Why is the color of bacteria in Figure 7C different from others, and why are there bacteria in the inhibition zones? The quality of the picture is very poor.

28. Line 329: a comma should be added after the word “defficidine”.

29. Line 334: “The non-cloned colony of bacteria slightly showed in wilt intensity as compared to control but lower than the cloned product.” That mean the non-cloned bacteria is better than cloned bacteria? “Lower” should be changed to “higher”?

30. Line 339: From table 5, I see the minimum growth of R. pseudosolanacearum was found at 50 µg/ml of a Macrolectin+ Difficidine+ Bacillaene (OD600=0.063) after 24 h. please check it.

31. Line 350: the word “own” means?

32. Line 350: format of the citation is wrong.

33. Line 355: a comma should be added between “macrolactin” and “difficidin”.

34. Line 357: the word “nobel” means?

35. Discussion: It seems that the results are mainly confirming what has previously been discovered in other studies, rather than highlighting what is new and relevant from their observations. Rewritten this part.

36. The quality of the figures is poor, and the figure legend should not be a picture.

Author Response

Reviewer 2

1 Line 11-17: Bacillus amyloliquefaciens DSBA-11 showed the best inhibition to R. pseudosolanacearum, and why use strain FZB42 to design primer sets? The rationale for why this species was selected is poorly defined.
      -We used nucleotide sequences of B. amyloliquefaciens strain FZB42 for designing the primers
        due to availability of whole genome sequence of FZB4 at NCBI database.
2 The abbreviations of strains should be uniform, and the name of genus is the full name when it first appears. some name of strains are nonstandard or misspelled, such as B subtilis on line 20.
          -Corrected in the manuscript as per suggested
3 Line19-21: How many strains of B. subtilis was used, they all possess difficidin and macrolactin. And only one strain MTCC-7092 belonging to B. pumilus was used?
         - Five strains of B. subtilis were used and they all posses difficidin and macrolactin. Yes, we use
          one strain of B. pumilus and B. licheniformis.
4 Line 25: Many type mistakes in the manuscript, such as comma after "bacillaene" should be not superscript. And replace "product" with "products".
        Corrected in the manuscript as per suggested
5 Capitalization should be uniform in Keywords.
        Corrected in the manuscript as per suggested
6 Line44: Some words are too colloquial, such as “very heavy” on line 44, “very good” on line 74-75.
     -  Corrected in the manuscript as per suggested
7 “Among them different species of Bacillus like B. amyloliquefaciens, B. subtilis, B. pumilus, B. coagulans, B. cereus, B. licheniformis, and B. vallismortis were applied for effective control of the disease [8, 9, 10, 7]. However, the Bacillus spp. have more advantages over other genera of bacterial antagonists,…”: There is no transition between the first sentence and the next sentence. The word "however" should be deleted.
         -  Corrected in the manuscript as per suggested
8 Line 57: The word "food" is inappropriate here, and "resources" may be more appropriate.
       -  Corrected in the manuscript as per suggested
9 Line 66: what the meaning of “co linearity”?
       - Colinearity occurs when amino acid sequences in the polypeptide corresponding to the codon
         sequences in nucleic acids, the 5′ end of the mRNA matching with the NH2 end of the
          polypeptide chain.
10 Line 68: delete “.” After “[19]”.
          - Corrected in the manuscript as per suggested
11 . Line 113: The name kits should be checked. Line 123: replace “data base” with “database”.
        - Corrected in the manuscript as per suggested
12 Line 133: a comma should be added before "1 ul".
         -  Corrected in the manuscript as per suggested
13 Line 136: annealing time "0.45 sec "is too short, "45 sec"?
        - Corrected in the manuscript as per suggested
14 Figure 1: Move Figure 1 to supplementary materials or delete.
           - The Figure 1 has been deleted.
15 Replace “such as” with “including” on line 184, 355
         - Corrected in the manuscript as per suggested.
16 Line 190: what the meaning is "old culture"? How to determine the concentration of the compound.
        - Corrected in the manuscript as per suggested.
17 The methods of 2.10 and 2.11.3 are repeated.
        - One set has been deleted.
18 Line 220: How to purify a gene? or you mean purify the product of polyketide? And usually “product of a gene” is protein. In here, the product may be polyketide, change the formulation of “product of gene”.
         - Corrected in the manuscript as per suggested
19 Line 248: The results of antagonistic ability of Bacillus spp. against R. pseudosolanacearum lacked statistical rigor.
      - Statistical analysis of data of Fig. 1 has been done and CD value has been mentioned in the text
          and also calculated standard deviation and standard error depicted in the Fig. 1.
20 Figure 2: The brackets in the ordinate are incomplete. Statistical analysis should be done between different species.
          - Corrected in Fig. 1 as earlier fig. 1 has been deleted.
21 Line 255-260: The genes macrolactin, and difficidin were also detected in DTBS-4 and DTBS-5. it is inappropriate to say the primers were specific for B. amyloliquefaciens.
         - In Bacillus subtilis, out of three, only two genes i.e.  macrolactin, and difficidin were amplified.
22 Line 262: change “up to” to “low to”?
         -  Corrected in the manuscript as per suggested
23 Figure 3 and 4: It is strange that in Figure 3, the DNA ladder is 100bp, while in Figure 4, it is 1kb. It seems that they are the same. Why there are 792, 705 and 616 bp bands in DNA marker. Marking the size of each stripe will make the results clearer.
         - In Fig. 3 & 4, 100bp ladder was used. Correction has been made in the legend of Fig. 4.
24 Line 299: Delete “cloning”.
        - Corrected in the manuscript as per suggested
25 Table 4: The error value of inhibition should be added.
         - Table 4, DMRT values are given against each value. There is no need to give error value.  
26 Figure 7: A control with no-clone is missing. Why is the color of bacteria in Figure 7C different from others, and why are there bacteria in the inhibition zones? The quality of the picture is very poor.
         - We took the photograph as actual showing the in the Petri plate. We missed taking
         photograph of non cloned product. We could not able to add photograph at this time.  Figure 7 c if
          required we can delete it.
27 Line 329: a comma should be added after the word “difficidin”.
        - Corrected in the manuscript as per suggested
28 Line 334: “The non-cloned colony of bacteria slightly showed in wilt intensity as compared to control but lower than the cloned product.” That mean the non-cloned bacteria is better than cloned bacteria? “Lower” should be changed to “higher”?
        - Corrected in the manuscript as per suggested.
29 Line 339: From table 5, I see the minimum growth of R. pseudosolanacearum was found at 50 µg/ml of a Macrolectin + Difficidin + Bacillaene (OD600=0.063) after 24 h. please check it. Checked it and corrected in the text.
30 Line 350: the word “own” means? Own has been deleted
31 Line 350: format of the citation is wrong. The correction has been done
32 Line 355: a comma should be added between “macrolactin” and “difficidin”.
- The correction has been done
33 Line 357: the word “nobel” means?
       - The word nobel has been deleted
34 Discussion: It seems that the results are mainly confirming what has previously been discovered in other studies, rather than highlighting what is new and relevant from their observations. Rewritten this part.
         -  Discussion has been revised as suggested.
35 The quality of the figures is poor, and the figure legend should not be a picture.
- Legends are separately mentioned in the text at last in the revised MS.

Reviewer 3 Report

This manuscript describes the detection of polyketide and its possible effect for Bacillus amyloliquefaciens DSBA-11 on the control of bacterial wilt. To emphasize the effect of specific polyketide on the disease control, the inhibitory effect against Ralstonia pseudosolanacearum were performed. Overall, this manuscript has scientific merit in discussing the contribution of polyketides. However, the research contents cannot meet the topic structure and the unreasonable experimental design are the issues for authors can think about. For example: If the authors want to use “mechanistic elucidation” as the main title, how to exclude other mechanisms that make polyketide as the most important factor is a required issue in this manuscript. Second, the use of TA cloning vector for gene expression in experimental design is unbelievable (Neither the position of the promoter nor the coverage of the open reading frame is mentioned).

Author Response

1 If the authors want to use “mechanistic elucidation” as the main title, how to exclude other mechanisms that make polyketide as the most important factor is a required issue in this manuscript.
            -  There are two objectives included in the MS, First is a screening of potential polyketide-
            producing strains of Bacillus species by using multiplex- PCR and the second is to study the
          mechanism of individual polyketide antibiotic particularly difficidin, macrolactin, and bacillaene.
             We highlighted the role of these polyketide antibiotics produced by Bacillus species.  Further
         we compared with cloned and noncloned product of these genes.
2 The use of TA cloning vector for gene expression in experimental design is unbelievable (Neither the position of the promoter nor the coverage of the open reading frame is mentioned).
          - We did not characterize all three polyketide antibiotic synthase genes. We cloned the
           genes as a described method in the MS. We used cloned product which was confirmed by
           colony PCR and used for further study. It requires further study of the structural and functional
          characterization of the genes.

Round 2

Reviewer 1 Report

Ms. was revised and can be accepted in the present form.

Author Response

Corrected MS as suggested by the reviewer in the text. 

Reviewer 2 Report

The authors have answered my comments and addressed my concerns. However, there are also some mistakes in the manuscript, please check it.

For example

Line200: "PowerSsoil" should be "PowerSoil"

Author Response

  1. Corrected in the text as PowerSoil and further checked the MS carefully 
  2. Improved the MS by adding some relevant references to discuss our result findings with the work of another scientist. 
  3.  The writing cloning methodology has been improved. 

Reviewer 3 Report

There are three points that cannot be assessed in the gene expression test:

1. The composition of one gene containing promoter and CDS. However, the gene expression in this study did not provide the gene is expressed by original promoter or T7/SP6 promoter in TA vector. According to the routine protein expression assays, the experiment require the use of expression vectors and further express the candidate protein in a specific E. coli strain (such as BL21). If only the TA vector is used, it is indeed difficult to believe this result. At the same time, according to the author's answer, it is impossible to judge that this experiment can be repeated by other research groups.

 2. If three candidate genes can be expressed in E. coli DH5a, it is still unknown whether it will harm E. coli cells.

3. How to quantify the polypeptides produced by Escherichia coli is not explained in the experiment. 2.11.1 is the analysis of polyketides, not the purification.

Author Response

  1. We used cloned products of polyketide antibiotic synthase genes using TA vector and confirmed them through colony PCR using specific primers for all three genes. we did not do a further study regarding sequencing, however, we tested their bioefficacy against bacterial pathogens. Further study is required as suggested by you. We are in the process but it could not be included in this paper.
  2.  We did not study the the cells of E. coli after cloning.  So we do not have any idea about the cells of E. coli. Your suggestion will be considered in a future study. 
  3.  Detail methodology of purification of the cloned product has been included in the Materials and methods on page no. 5 item  2.11.1 as suggested.